Classification of RNA backbone conformations into rotamers using 13C′ chemical shifts: exploring how far we can go

Icazatti Alejandro A. ale.icazatti@gmail.com 1
Loyola Juan M. 1
Szleifer Igal 2 3 4
Vila Jorge A. 1
Martin Osvaldo A. 1
1 IMASL - CONICET, Universidad Nacional de San Luis , San Luis , Argentina
2 Department of Biomedical Engineering, Northwestern University , Evanston , IL , United States of America
3 Chemistry of Life Processes Institute, Northwestern University , Evanston , IL , United States of America
4 Department of Chemistry, Northwestern University , Evanston , IL , United States of America
Muhle-Goll Claudia
Electronic publication date: 2019 Oct 21
Publication date: 2019
Volume: 7
Electronic Location ID: e7904
Received 2019 Jun 3; Accepted 2019 Sep 16
Copyright: ©2019 Icazatti et al.
Copyright year: 2019
Copyright holder: Icazatti et al.
License: This is an open access article distributed under the terms of the Creative Commons Attribution License, which permits unrestricted use, distribution, reproduction and adaptation in any medium and for any purpose provided that it is properly attributed. For attribution, the original author(s), title, publication source (PeerJ) and either DOI or URL of the article must be cited.
License URL: https://creativecommons.org/licenses/by/4.0/

Keywords: RNA, Rotamers, Machine learning, Chemical shifts, DFT

Funding: Consejo Nacional de Investigaciones Científicas y Técnicas (Argentina) PIP-0087 Agencia Nacional de Promoción Científica y Tecnológica (Argentina) PICT-0556 PICT-0218 This research was supported by grants from: Consejo Nacional de Investigaciones Científicas y Técnicas (Argentina) (PIP-0087 to Jorge Alberto Vila) and Agencia Nacional de Promoción Científica y Tecnológica (Argentina) (PICT-0556 to Jorge Alberto Vila, PICT-0767 to Jorge Alberto Vila and PICT-0218 to Osvaldo Antonio Martin). The funders had no role in study design, data collection and analysis, decision to publish, or preparation of the manuscript.

==============================
The conformational space of the ribose-phosphate backbone is very complex as it is defined in terms of six torsional angles. To help delimit the RNA backbone conformational preferences, 46 rotamers have been defined in terms of these torsional angles. In the present work, we use the ribose experimental and theoretical 13C′ chemical shifts data and machine learning methods to classify RNA backbone conformations into rotamers and families of rotamers. We show to what extent the experimental 13C′ chemical shifts can be used to identify rotamers and discuss some problem with the theoretical computations of 13C′ chemical shifts.

Introduction

Nucleic acids are central macromolecules for the storing, flow and regulation of genetic and epigenetic information in cellular organisms. RNA can adopt a wide variety of 3D structural conformations and this structural variability explains the multiplicity of roles that RNA performs on cells (Wan et al., 2011; Eddy, 2001). The classification of RNA backbone conformations into rotamers is a very useful way to delimit the conformational space of RNA structures. Rotamers are defined in terms of the backbone torsional angles namely α, β, γ, δ, ε and ζ (as shown in Fig. 1). This classification was proposed by Richardson et al. (2008), and has been achieved after the attempts of different research groups to find a consensus RNA backbone structural classification. There are 55 backbone rotamers, from which 46 are rotamers with well defined torsional angles distributions, and the remaining nine rotamers were proposed as wannabe rotamers. The ‘suite’ is the basic subunit used for rotamer classification. The suite is defined from sugar-to-sugar (or from the δ torsional angle of residue i-1 to the δ torsional angle of residue i), and it is contained within the dinucleotide (DN) subunit (see Fig. 1). 13C′ chemical shifts (CS) have been successfully used by our and other groups for proteins and glycans structural determination, validation and refinement (Shen & Bax, 2010; Martin et al., 2013; Frank et al., 2015; Garay & Vila, 2018). 1H CS have been successfully used by Sripakdeevong et al. (2014) for structure determination and prediction of noncanonical RNA motifs. Methods incorporating 13C CS for RNA structure determination, validation and refinement are also available (Frank, Stelzer & Bae, 2013; Frank, Law & Brooks, 2014; Brown, Summers & Johnson, 2015) but, to our knowledge, none of them include the explicit use of backbone rotamers. In this work, we study how to use 13C′ CS to classify RNA backbone conformations into rotamers with machine learning models. Overall, a complete understanding of the molecular basis of the biological processes in which RNA molecules are involved entails an accurate knowledge of their 3D structure. In this regard, it is well known that the computation of the 13C′ chemical shifts (CS) for RNA, at DFT-level of theory, is very sensitive to the backbone conformation of the molecule. Thus, among other potential application of our work is to build, for any possible combination of RNA backbone torsional-angles conformations, a detailed 13C CS look-up table. Hence, given a 13C CS value the corresponding set of RNA backbone torsional angles can be quickly determined, and vice versa, making the look-up table a very valuable tool with which determine, validate and refine RNA conformations.

Figure 1 RNA DN template with sequence AA, obtained from a random PDB structure.

C, H, O, N and P nuclei are colored in green, white, magenta, blue and yellow, respectively. Torsional angles of RNA backbone are named on Greek characters (α, β, γ, δ, ε, ζ). Suite (from δi−1 to δi), DN and nucleotide subunits are indicated.

Methods

In order to provide a clear understanding of the methodology implemented in this work, a flowchart with the overall work-flow is shown in Fig. 2. A theoretical dataset of RNA backbone rotamers with 13C′ CS values is necessary to train the machine learning models to classify RNA experimental suites into rotamers. In the following two sections we explain how we obtained both datasets.

Figure 2 Flowchart of the general work-flow followed in this work.

The experimental data retrieval process and the theoretical data generation steps are indicated inside the green and the blue boxes, respectively. The classification step using machine learning models is indicated with an orange box. The term rotamers could indicate the original backbone rotamers or redefined rotamer families.

Experimental dataset

Experimental 13C′ CS data for RNA molecules was retrieved from the BioMagResBank (BMRB; http://www.bmrb.wisc.edu) (Ulrich et al., 2008), along with their corresponding structures from the Protein Data Bank (PDB; https://www.rcsb.org/) (Berman et al., 2000). As it is fundamental to count on reliable experimental 13C′ CS values for an accurate structural analysis, data curation was carried out using 13Check_RNA (Icazatti et al., 2018) a Python module to correct RNA 13C′ CS systematic errors, recently developed in our group. The obtained dataset (see Table S1) contains 26 RNA structures with 13C′ CS for the five ribose carbon nuclei (C1′, C2′, C3′, C4′ and C5′), providing a total of 391 suite subunits and 391 sets of 13C′ CS. As there are at least 8 models in the NMR ensembles for each RNA molecule (up to 20 in some cases), the complete database contains 7,612 conformations. Given that we needed a one-to-one correspondence between the sets of CS and the rotamer suites, only the first structure from each NMR ensemble was used, considering that the first model listed in the PDB files is usually reported as the structure with the lowest energy scoring. This choice has a negligible average effect on the results of our analysis (see Figs. S10 and S11).

For every PDB entry, the 3D coordinates of the first model were extracted in order to compute the backbone torsional angles (δi−1, εi−1, ζi−1, αi, βi, γi, δi) of the suites. Then, these torsional angles were used to assign the RNA suites to their corresponding rotamer names. From the 46 original rotamers, only 38 are represented in the final experimental dataset.

Theoretical dataset

In order to have a complete dataset with the 46 RNA backbone rotamers and their corresponding 13C′ CS, a theoretical dataset was also constructed. A template for each of the 16 possible combinations of DN (A, C, U and G) sequences was obtained from RNA structures found in the PDB. A Monte-Carlo conformational sampling was carried out by rotating the backbone torsional angles of the corresponding suite contained in each DN, while keeping the bond-lengths and bond-angles fixed (rigid geometry approximation). To perform such rotations, the torsional angle distributions for each of the 46 RNA backbone rotamer suites (Richardson et al., 2008) were used. A function which eliminates conformers with atom clashes was implemented as part of the routine. As a result, 10,340,852 conformations were generated. Quantum-theory level computation of CS is very time-consuming. Therefore, to reduce the number of calculations, a smaller number of conformations was selected. Aiming to keep most of the variability of the originally generated conformations, we computed the Shannon entropy S (see Eq. (1)) of the distribution of torsional angles. Here, Pi is the probability of the i conformation taken from a histogram with a bin size of 5 degrees. The entropy was computed for different subsets of conformations and sample sizes (from 5 to 100) (see Fig. 3). We decided to use the 80% of the maximum entropy as a cutoff, which implies a sample size of around 40 conformations per rotamer. As we also considered the 16 combinations of DN sequences, the total number of conformations computed at the DFT level of theory was 30,530. (1) S=−∑iPi lnPi

Figure 3 Percentage of entropy of the sample against sample size for a given DN sequence and rotamer, UU and 1a, respectively, in this case.

The red line and the blue bars represent the mean and the range of percentage of entropy for a given sample size, respectively.

Details of the quantum-chemical calculations of the 13C′ shieldings

Previous to the DFT calculations of the obtained dataset, a test was performed over a subset of 41 rotamers of sequence AA. A similar approach as described below for mononucleotides was used, except that the templates were methyl-blocked DNs: Me − O3′i−2 − Ai−1 − Ai − O5′i+1 − Me. Subsequent comparison of the obtained 13C′ CS for these DNs with those obtained from the corresponding mononucleotides, gave the same result within 10−2 ppm while the total computation time was approximately half the total time for computing the complete DNs. Thus, the DN conformations from the final dataset were split in their corresponding mononucleotide subunits. Nucleotide subunits were treated as terminally-blocked mononucleotides with methyl groups (Me) in both termini (Me − O3′i−1 − Xi − O5′i+1 − Me). Phosphate groups of the backbone were treated as neutral, because we assume that all backbone charges are shielded during the quantum-chemical calculations. Results based on the analysis of 139 conformations of ubiquitin at pH 6.5 (Vila & Scheraga, 2008), indicate that use of neutral, rather than charged, aminoacids is a significantly better approximation of the observed 13Cα CS in solution for the acidic groups, and a slightly better representation, though significantly less expensive in computational time, for the basic groups. Considering that the phosphate group in RNA is close to the nucleus of interest (as it happens with the acidic groups) we can assume, without losing generality, that neutral rather than charged phosphate group is a better approximation for the computation of the 13C′ CS in the RNA suites. This approach was also adopted because under physiological conditions, the phosphate groups are completely ionized and neutralized by positive charges (Lehninger Nelson & Cox, 2000). A 6–311+G(2d,p) locally dense basis set (Chesnut & Moore, 1989) was used for calculation of backbone 13C′ CS and their nearest neighbor nuclei, at the DFT level of approximation (see Fig. 4 for details). The remaining nuclei were treated with a 3-21G basis set. The OB98 density functional was used because good results were previously observed for proteins and glycans in our group (Vila & Scheraga, 2009; Garay et al., 2014). All DFT computations were done using the Gaussian package (Frisch et al., 2004). Summarizing, the adopted strategies make the computed 13C′ CS from mononucleotides suitable for comparison with the 13C′ CS observed from complete RNA molecules.

Figure 4 Example of a methyl blocked mononucleotide used for DFT calculations.

The locally-dense basis-set approach is indicated by the different colors: the nuclei in magenta were treated with the extended 6-311+G(2d,p) basis set and the nuclei in green were treated with the smaller 3-21G basis set.

Figure 5 Distribution plots for the six RNA backbone torsional angles α, β, γ, δ, ε and ζ in (A), (B), (C), (D), (E) and (F), respectively.

Torsional angles values were obtained from the RNA09 database used in Murray LW. 2007. RNA Backbone Rotamers and Chiropraxis. Doctoral Dissertation, Dept. of Biochemistry, Duke University, Durham, NC, USA.

Table 1 The 46 RNA backbone rotamers were arranged in 22, 10, 10, 7 and 4 families of rotamers based on the observed distributions of δi−1δiαγ, δi−1δiα, δi−1δiγ, αγ and δi−1δi torsional angles values, respectively.

Additionally, the 46 rotamers were separated in RNA A–form helix vs. no A–form helix rotamers in two ways: (i) RNA A-form helix rotamer 1a vs. the remaining no A–form helix rotamers (A_noA families) and (ii) rotamers related to A–form helix (i.e., 1a, 3d, 3b, 5d, 0a, 6b, 4b) vs. the remaining rotamers (A*_noA* families).

46 rotamers	22 families δi−1δiαγ	10 families δi−1δiα	10 families δi−1δiγ	7 families αγ	4 families δi−1δi	2 families A_noAi	2 families A*_noA*ii	
&a	e	a	a	e	a	b	b	
#a	q	c	c	e	c	b	b	
0a	q	c	c	e	c	b	a	
0b	t	d	d	e	d	b	b	
0i	o	g	g	b	c	b	b	
1[	l	b	b	e	b	b	b	
1a	e	a	a	e	a	a	a	
1b	l	b	b	e	b	b	b	
1c	d	e	e	d	a	b	b	
1e	f	e	e	f	a	b	b	
1f	d	e	e	d	a	b	b	
1g	c	a	a	c	a	b	b	
1L	e	a	a	e	a	b	b	
1m	e	a	a	e	a	b	b	
1o	m	i	i	g	b	b	b	
1t	k	f	f	d	b	b	b	
1z	j	b	b	c	b	b	b	
2[	t	d	d	e	d	b	b	
2a	q	c	c	e	c	b	b	
2h	r	g	g	f	c	b	b	
2o	v	j	j	g	d	b	b	
3a	e	a	a	e	a	b	b	
3b	l	b	b	e	b	b	a	
3d	a	a	a	a	a	b	a	
4a	q	c	c	e	c	b	b	
4b	t	d	d	e	d	b	a	
4d	n	c	c	a	c	b	b	
4g	p	c	c	c	c	b	b	
4n	o	g	g	b	c	b	b	
4p	s	d	d	a	d	b	b	
4s	u	h	h	f	d	b	b	
5d	a	a	a	a	a	b	a	
5j	b	e	e	b	a	b	b	
5q	h	f	f	b	b	b	b	
5z	j	b	b	c	b	b	b	
6d	n	c	c	a	c	b	a	
6g	p	c	c	c	c	b	b	
6j	o	g	g	b	c	b	b	
6n	o	g	g	b	c	b	b	
6p	s	d	d	a	d	b	b	
7a	e	a	a	e	a	b	b	
7d	a	a	a	a	a	b	b	
7p	g	b	b	a	b	b	b	
7r	i	i	i	c	b	b	b	
8d	n	c	c	a	c	b	b	
9a	e	a	a	e	a	b	b	

Families of rotamers

The original 46 RNA backbone rotamers were grouped in families based on their δi−1, δi, α and γ torsional angles values. Only these four (out of seven) backbone torsional angles in the suite subunit were chosen to group the rotamers because their distributions of observed values are bimodal (δi−1 and δi) and trimodal (γ and α), with clearly separated peaks (see Fig. 5). This selection allowed us to group rotamers based on the torsional angle values within the different peaks. As summarized in Table 1, four families were found when both δi−1 and δi torsional angles in the suite were used (see Table 2), seven families for the αγ combination, 10 families for δi−1δiα, and δi−1δiγ, and 22 families for δi−1δiαγ. In order to evaluate the classification performance of the RNA A–form helix conformations, the rotamers were also grouped as: (i) A_noA families, where the 46 rotamers were separated in A–form helix (1a) vs. no A–form helix rotamers, and (ii) A*_noA* families, where the 46 rotamers were separated in rotamers related to A–form helix (1a, 3d, 3b, 5d, 0a, 6b and 4b rotamers) vs. the remaining rotamers.

Classification

A series of machine learning methods were used to classify RNA suites as rotamers (or families of rotamers) based on their ribose 13C′ CS values. The following classification methods from the scikit-learn Python library (Pedregosa et al., 2011) were trained: K-Nearest Neighbors (NN), Decision Tree (DT), Random Forest (RF), Support Vector Machine (SVM) and a class of neural network called Multi-Layer Perceptron (MLP). Different model parameters were tried out (see Table S3). A random sampling algorithm was also used as a control, where suites were classified randomly. The sequence of the suite was considered for classification, because we found that the performance increased compared to a sequence–independent classification (see Fig. S10). The classification performance was assessed with four measures: weighted accuracy, precision, recall and F1 score (Van Rijsbergen, 1979). The weighted accuracy was used in order to recalibrate the contribution of the different rotamers, because the observed frequency of the rotamers is highly uneven (see Fig. S1). The weights used in the weighted accuracy were obtained from a substitution matrix (ROSUM, for ROtamers SUbstitution Matrix). The definition of the ROSUM matrix was inspired by the BLOSUM matrix used for protein sequence alignment (Henikoff & Henikoff, 1992). The matrix is used to weight the match or no match, between the true rotamer and the predicted rotamer, as a function of the euclidean distance between rotamers (in the seven-dimensional space of the suite backbone torsional angles) and the observed frequency of each rotamer. The torsional angles values and the observed frequencies are extracted from the rotamers table (Richardson et al., 2008). A ROSUM matrix was obtained for each of the rotamer families described in the previous section. Further details on the construction of the ROSUM matrices are provided in Data Section S4. The precision and recall were used because they gave a general overview of the performance of the method. In particular, they allowed us to assess the fraction of classified items that were correctly identified and the sensitivity of the method. The F1 score was also used as a performance measure because it is the harmonic mean of precision and recall and as such, it gives more realistic measure of the classifier’s performance.

Table 2 Mean torsional angles values of the representative (i.e., most frequent) rotamers from the four δi−1δi families.

Values were extracted from the rotamer table of Richardson et al. (2008).

δi−1δi families	46 rotamers	δ(i−1)	ε(i−1)	ζ(i−1)	α(i)	β(i)	γ(i)	δ(i)	
a	1a	81	212	289	295	174	54	81	
b	1b	84	215	289	300	177	58	145	
c	2a	145	260	289	288	193	53	84	
d	2[	146	259	291	292	210	54	148	

Experimental vs. theoretical

The classification models trained with theoretical data were used to classify the experimental suites. The result of the theoretical calculations (described in a previous section) are theoretical NMR isotropic shieldings (σ). The theoretical shieldings (σcomp) must be subtracted from a reference shielding value (σref) to be transformed into theoretical CS (δcomp) (see Eq. (2)) which can then be compared with the experimental CS (δexp). A simple reference value of σref = 185.00 ppm was used, which is very close to the theoretical isotropic shielding for TMS (σTMS,th) (Vila & Scheraga, 2009), and it is consistent with the reference value previously defined for proteins and glycans. Alternatively, a set of effective references were obtained as a function of: (i) the nitrogenous base sequence, (ii) the combinations of ribose puckering states in the four families of rotamers obtained from δi−1δi torsional angles distributions, (iii) the five carbon nuclei 13C′ CS mean values and (iv) a linear regression between theoretical and experimental ribose 13C′ CS values for a set of suites (see Table S2). (2) δcomp=σref−σcomp.

Theoretical vs. theoretical

The classification models trained with theoretical data were also used to classify the theoretical suites. In this case, classification was assessed through a leave-one-out cross-validation (LOO-CV). In LOO-CV, the dataset is split into a test set and training set in a one-folded manner, which means that at every iteration a unique suite is taken apart from the dataset and the remaining suites are used for training. This process continues until every suite from the theoretical dataset is evaluated.

Experimental vs. experimental

A LOO-CV was also used to classify the experimental suites.

Results and Discussion

For experimental vs. theoretical classification (Fig. 6) the 46 rotamers can be classified by means of backbone 13C′ CS with a maximal F1 score of 0.34 (see Table S5). When the 46 rotamers are grouped in families based on their torsional angles distributions, the highest scores correspond to the use of δ(i−1) and δ(i) torsional angles, where all the classifiers gave maximal scores above 0.65. This result is in agreement with the fact that backbone 13C′ CS are highly sensitive to ribose puckering states (Giessner-Prettre & Pullman, 1987), since the δ torsional angle keeps a direct relation with the ribose puckering (Gelbin et al., 1996). The δi−1δiγ, δi−1δiα, δi−1δiαγ and αγ families also show improved scores over the classification of the 46 rotamers. The A*_noA* and A_noA families show low classification scores relative to their random choice classification scores, which means that backbone 13C′ CS cannot distinguish between A–form helix and no A–form helix rotamers. In general the use of more complex classifier models such as Neural Networks, Support Vector Machine, Decision Tree and Random Forest does not assure a better performance for the current task, thus the simpler Nearest Neighbor model can be chosen for classification into RNA rotamers. In both the theoretical vs. theoretical and the experimental vs. experimental classifications (see Figs. 7 and 8, respectively), the performances increase for every group of families, compared to the experimental vs. theoretical classification. In the theoretical vs theoretical classification the performance values are very close to 1.0 for δi−1δi families and A–form helix vs. no A–form helix rotamers (A_noA). In the theoretical vs. theoretical classification, the performance value ranges are particularly narrow, except for MLP and SVM classifiers.

Figure 6 Box-plots with the weighted accuracy and F1 score for the experimental vs. theoretical classification of rotamers and families of rotamers, using Nearest Neighbor (NN), Decision Tree (DT), Random Forest (RF), Multi-Layer Perceptron (MLP) and Support Vector Machine (SVM) classifiers.

A random-choice (RAND) algorithm was used as a baseline reference. Classification results for the 46, δi−1δiαγ, δi−1δiα, δi−1δiγ, αγ and δi−1δi, A*_noA* and A_noA rotamer families are shown in (A), (B), (C), (D), (E), (F), (G) and (H) respectively. The highest values of weighted accuracy and F1 score, for the experimental vs. theoretical classification along with parameters of the classifiers are provided in Tables S4 and Fig. S1. Precision and recall are shown in Fig. S12.

Figure 7 Box-plots with the weighted accuracy and F1 score for the theoretical vs. theoretical classification of rotamers and families of rotamers, using Nearest Neighbor (NN), Decision Tree (DT), Random Forest (RF), Multi-Layer Perceptron (MLP) and Support Vector Machine (SVM) classifiers.

A random-choice (RAND) algorithm was used as a baseline reference. Classification results for the 46, δi−1δiαγ, δi−1δiα, δi−1δiγ, αγ and δi−1δi, A*_noA* and A_noA rotamer families are shown in (A), (B), (C), (D), (E), (F), (G) and (H) respectively.

Figure 8 Box-plots with the weighted accuracy and F1 score for the experimental vs. experimental classification of rotamers and families of rotamers, using Nearest Neighbor (NN), Decision Tree (DT), Random Forest (RF), Multi-Layer Perceptron (MLP) and Support Vector Machine (SVM) classifiers.

A random-choice (RAND) algorithm was used as a baseline reference. Classification results for the 46, δi−1δiαγ, δi−1δiα, δi−1δiγ, αγ and δi−1δi, A*_noA* and A_noA rotamer families are shown in (A), (B), (C), (D), (E), (F), (G) and (H) respectively.

The high scores obtained for the theoretical vs. theoretical classification indicates that 13C′ CS are in fact very sensitive to changes of the torsional angles, the only variable we changed for the construction of the theoretical dataset. At the same time the lower performance obtained in the experimental vs. theoretical classification, is signalling that the atomistic model used for the DFT computations is not good enough to reproduce the experimental observations.

One reason the theoretical vs. theoretical classification gives better results compared to both the experimental vs. experimental and the experimental vs. theoretical classifications, could be that the experimental database is very sparse and the theoretical dataset is instead dense, or in other words the coverage of the theoretical dataset is much better than the experimental one. To explore if this is in fact a reasonable explanation, we removed elements from the theoretical dataset to mimic the sparsity of the experimental dataset (see Fig. S13). We found that while the weighted accuracy decreased (on average 0.09 points) this is not enough to explain the lower performance of the experimental vs. theoretical (on average 0.31 points lower) or experimental vs. experimental (on average 0.16 points lower) classifications. In another experiment, noise on the order of the expected error (1.47 ppm) between experimental and theoretical 13C′ CS for those rotamers correctly classified, was added to the theoretical 13C′ CS and then a theoretical vs. theoretical + noise classification was performed (see Fig. S14). Both tests reinforce the idea discussed in the previous paragraph, i.e., we need a better model for the theoretical DFT computations. These experiments also provide indirect evidence indicating that the accuracy of the experimental vs. experimental classification will be improved as more RNA conformations are deposited in databases giving another incentive to determine and deposit RNA structures and 13C′ CS data.

Conclusion

In this work, we explored the use of RNA backbone 13C′ CS to classify backbone conformations into rotamers and families of rotamers. In general, our study led us to the following conclusions: (1) the classification of the rotamer families defined by the δ torsional angles (see Table 2), which are directly related to the ribose puckering states, gives the best performances, in line with the results previously described by other authors; (2) classification of A-form helix and no A-form helix rotamers using 13C′ CS is not better than a random classification; (3) the performance achieved using the simple Nearest-Neighbor method is on a par with more complex classifiers such as Neural Networks, Support Vector Machine, Decision Tree and Random Forest; (4) 13C′ CS values are able to sense changes in torsional angles, but they are also affected by other factors, thus future DFT computations of RNA 13C′ CS should use more complex models than the one used in this work; (5) experimental 13C′ CS can be useful to identify RNA rotamers, if the rotamers are re-grouped in smaller families as the 46 rotamers seems to be a too fine description for accurate discrimination in terms of 13C′ CS; (6) the usefulness of 13C′ CS for rotamers identification should improve as more RNA structures and experimental 13C′ CS become available.

Supplemental Information

Supplemental Information 1 Supplementary Data

Click here for additional data file.

We greatly appreciate Myriam Villegas for valuable discussions, comments and suggestions.

Additional Information and Declarations

Competing Interests

Author Contributions

Data Availability

The authors declare there are no competing interests.

Alejandro A. Icazatti conceived and designed the experiments, performed the experiments, analyzed the data, prepared figures and/or tables, authored or reviewed drafts of the paper, approved the final draft.

Juan M. Loyola conceived and designed the experiments, performed the experiments, analyzed the data, contributed reagents/materials/analysis tools, authored or reviewed drafts of the paper, approved the final draft.

Igal Szleifer analyzed the data, authored or reviewed drafts of the paper, approved the final draft.

Jorge A. Vila conceived and designed the experiments, analyzed the data, contributed reagents/materials/analysis tools, authored or reviewed drafts of the paper, approved the final draft.

Osvaldo A. Martin conceived and designed the experiments, analyzed the data, contributed reagents/materials/analysis tools, authored or reviewed drafts of the paper, approved the final draft.

The following information was supplied regarding data availability:

Data is available at https://github.com/BIOS-IMASL/RNA_13C_ChemicalShifts_Databases.

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
