# Peer review of "Classification of RNA backbone conformations into rotamers using 13C′ chemical shifts: exploring how far we can go"

_PeerJ, doi:10.7717/peerj.7904_

## Round 0.1 · original submission · Minor Revisions

The manuscript is well written but uses mainly highly specialized technical terms. Can you add a few lines for less experienced readers how to best use the information derived from 13C' CS to predict backbone conformations - e.g. which method to use?
Figure 6 should be revised using larger fonts. E.g. a),b),c) and headlines are hardly visible at first sight.

Reviewer 1 ·

Basic reporting

1) No issues with writing.

2) In line 35-36, the authors should consider citing the work of Das and coworkers: ‘Structure determination of noncanonical RNA motifs guided by 1H NMR chemical shifts’, which describes the use of 1H chemical shifts to guide determine/prediction of RNA structure.

3) Figure 5 was reproduced from a dissertation, and though it was properly cited, the authors should consider producing their own version of this figure.

4) Raw data, QM input scripts, and machine learning scripts should be made available (currently missing).

Experimental design

1) Original primary research within Aims and Scope of the journal?

Yes. RNAs play importance biologically roles and to understand how they function, we must understand their structure.

2) Research question well defined, relevant & meaningful. It is stated how research fills an identified knowledge gap?

Yes. The research aimed to assess the extent to which 13C chemical shifts can be used to classify RNA rotamer states. Answer to this question could help develop methods to maximize the amount of structural information about an RNA that can be extracted from NMR chemical shift data.

3) Rigorous investigation performed to a high technical & ethical standard?
Yes.

4) Methods described with sufficient detail & information to replicate?
Yes.

Validity of the findings

No comment

Additional comments

In line 35-36, the authors should consider citing the work of Das: ‘Structure determination of noncanonical RNA motifs guided by 1H NMR chemical shifts’, which the describes the use of 1H chemical to guide determine/prediction of RNA structure.

When describing the experimental datasets, the authors should include the number of conformations as well as the total number of set of {C1’,C2’,C3’,C4’C5’} CS data

For the DFT calculations, much of the justifications of the methods used are based on calculations carried out proteins. The authors should explore work by Case and coworkers to get some guidance on choosing of DFT functionals and basis set for computed NMR shielding in nucleic acids. For example, see AFNMR: automated fragmentation quantum mechanical calculation of NMR chemical shifts for biomolecules, and the references therein.

The impact of using the 3-21G basis set for some of the nuclei and extended 6-311+G(2d,p) basis set for the result is not fully discussed or should be justified by showing that the added computational expense of using the extended 6-311+G(2d,p) basis set for all nuclei does not yield significantly different results.

The lack of good performance for the experimental vs. theoretical, compared to theoretical vs. theoretical, was attributed to problems with atomistic models. Another reason for the poor relative performance is inherent errors in the computed chemical shifts. To test this, authors should consider adding noise to the theoretical database and then carrying out a theoretical vs theoretical+noise analysis. By adding noise on the order of the expected errors between actual and theoretical chemical shifts, they will be able to test how errors in the computed shifts affect the performance of the classifiers.

As noted by the authors, the experimental database used to train the classifiers was sparse, compared to the theoretical database. The author should consider using an imputation strategy to impute missing 13C ribose chemical shifts from existing 13C ribose chemical shifts, and thus, increase the number of RNA in their dataset. Currently, there are > 70 RNAs for which both 13C chemical shifts and NMR structures are available in the BMRB and PDB, respectively.

Raw data such as coordinate files, and QM calculation and machine learning scripts should be made available (currently missing).

Reviewer 2 ·

Basic reporting

The text is well written in very good English; some minor issues, which are likely to be typos, pointed out in the comments to the authors. Literature references are adequate. Article structure and the form of tables and figures have no issues. The results presented support the conclusions; however, a table/figure with the geometric parameters/pictures of sample structures from the families dinucleotide conformations considered in this work would be desirable.

Experimental design

No issues in this part. The paper fits to the Aims and Scope of the journal, the research questions are well defined and adequately addressed. The methods used are relevant and explained sufficiently.

Validity of the findings

A table/figure with the geometric parameters/pictures of sample structures from the families dinucleotide conformations considered in this work would be desirable for readers' benefit.

Additional comments

This is a good paper which undertakes a difficult taks of the classification of RNA backbone conformations based on 13C NMR chemical shifts. The Authors divided the conformational space of dinucleotides into 46 possible conformations (based on the backbone angles) of which 38 were found in the experimental structures.. The conforomers were taken from the experimental structures and generated from the dihedral-angle distribution. Subsequently, the 13C chemical shifts were computed and correlated with conformer geometry to determine families using a menu of machine-learning methods. Careful statistical assessment of the validity of the resulting classification was done. The delta(i-1) and delta(i) angles related to the puckering of the flanking ribose rings were found to be the best descriptors that could be translated inito chemical shifts. Interestingly, the global-fold properties (A-non A helix, A-helix-everytning else) were found not to be related to the chemical shifts.

In my opinion this paper contains very useful results, because the classification enalbles a researcher to estimate the type of backbone conformation just by measuring 13C chemical shitt, which is relatively simple. I have a couple of minor suggestions:

1. A picture with the representatives of the 4 families of conformers classified based on delta(i-1) and delta(i) summarized in Table 1 or a table with the dihedral angles corresponding to those representative would be useful.

2. Were the theoretically-generated conformers energy-minimized? If so, by what method? It not, was the set of generated conformers pruned to eliminate those with atom clases?

3. page 4, lines 93-95:
"Considering that the phosphate group in RNA is close to the nucleus of interest (as it happens with the
94 acidic groups) we can assume, without losing generality, that neutral rather than charged phosphate group
is a better approximation for the computation of the 13C′ 95 CS in the RNA suites."
The general conclusion is correct but the rationale of using protonated phosphate groups is rather that they are surrounded by counter ions (mainly the sodium ions). The phosphate group is rather unlikely to be protonated at physiological conditions.

4. I found a couple of minor grammar errors, e.g.:page 4, lines 89-91 (corrections in caps):

"Results based on the analysis of 139 conformations of ubiquitin
90 at pH 6.5 (Vila and Scheraga, 2008), INDICATE that use of neutral, rather than charged, amino acids is a
91 significantLY better approximation of the observed 13Ca CS in solution for the acidic groups"

---

## Round 0.2 · accepted · Accept

Reviewer 1 noted that overall the responses to their comments were adequate and the manuscript is much improved. As such, I recommend that the manuscript be published as is.

Reviewer 1 ·

Basic reporting

No comment

Experimental design

No comment

Validity of the findings

No comment

Additional comments

No comment